# Crystal nuclei templated nanostructured membranes prepared by solvent crystallization and polymer migration

Bo Wang[1], Jing Ji[1] & Kang Li[1]

Currently, production of porous polymeric membranes for filtration is predominated by the phase-separation process. However, this method has reached its technological limit, and there have been no significant breakthrough over the last decade. Here we show, using polyvinylidene fluoride as a sample polymer, a new concept of membrane manufacturing by combining oriented green solvent crystallization and polymer migration is able to obtain high performance membranes with pure water permeation flux substantially higher than those with similar pore size prepared by conventional phase-separation processes. The new manufacturing procedure is governed by fewer operating parameters and is, thus, easier to control with reproducible results. Apart from the high water permeation flux, the prepared membranes also show excellent stable flux after fouling and superior mechanical properties of high pressure load and better abrasion resistance. These findings demonstrate the promise of a new concept for green manufacturing nanostructured polymeric membranes with high performances.

[1] Department of Chemical Engineering, Imperial College London, London SW7 2AZ, UK. Correspondence and requests for materials should be addressed to K.L. (email: kang.li@imperial.ac.uk).

Filtration is a separation process based on size exclusion through a porous media, where liquids and small particles pass the pores, but bigger particles are rejected. Porous membranes have been widely used in liquid filtration for drinking water production, wastewater treatment, dialysis, beverage clarification and so on. Membrane-based filtration is now a business of tens of billions USD per year, among which microfiltration (pore size > 100 nm) and ultrafiltration (pore size ranging from 2 to 100 nm) share the biggest part of the total membrane market. Among all types of microfiltration/ultrafiltration membrane materials, polyvinylidene fluoride (PVDF) is one of the most commonly used membrane materials because of its outstanding properties such as inertness in a wide range of harsh chemical and thermal conditions, particularly surviving from chlorination disinfection with excellent mechanical strength in working conditions[1,2], making it predominant in the pre-treatment units of seawater desalination and in wastewater treatment. Nevertheless, PVDF membranes suffer from low permeation fluxes, and most commercial ultrafiltration PVDF membranes for industrial use only possess pure water permeation flux of less than 200 litres per square meter membrane area per hour (LMH) under 1 bar pressure difference across the membrane. To compensate the low flux of PVDF membranes, larger membrane areas are required to treat a large volume of water. It is very often that in a seawater desalination plant, the pre-treatment unit composed of hundreds of PVDF ultrafiltration membrane module trains occupies a large footprint, and the total PVDF membrane area could exceed 200,000 $m^2$ for a desalination plant with a capacity of 100,000 $m^3$ per day. The requirement of large membrane area increases not only the capital investment, but also the daily operating costs (for energy and maintenance) of the filtration units. Therefore, improving the permeation flux of the PVDF membranes is crucial to reduce the costs and energy consumption in filtration plants. Currently, PVDF membranes as well as other microfiltration/ultrafiltration polymeric membranes are produced via phase-separation methods[3–5], predominately the non-solvent induced phase-separation (NIPS) method[2,5], although some commercial membranes are also produced by the thermal-induced phase-separation (TIPS) method[6–8]. An excellent review on the preparation and modification of PVDF membranes has been provided by Liu et al.[2] Complex physical–chemical factors are involved in the NIPS process, such as inter-diffusion of solvent and non-solvent, rheology of polymer solution, interfacial instabilities and even ambient temperature and humidity[2,5,9]. Therefore controlling the quality of final membrane products is extremely complicated, and often an ideal structure with minimized permeation resistance is difficult to achieve.

In production of porous materials, a simple approach of freeze drying is often used[10–12]. This method utilizes randomly oriented solvent crystallites as template to produce flow passages (or pores) of micron-scale size in the porous materials, where the separation takes place normally via adsorption rather than size exclusion. A few attempts have been made to produce membranes via this approach. However, owing to the lack of effective control of the size and orientation of the solvent crystallites, those attempts failed to produce membranes with pore size smaller than 250 nm (refs 13,14), which is no difference compared with the flow passage of porous materials prepared by the same technique.

In practice of the freeze drying approach, the actual nuclei/crystallite sizes obtained are determined by the kinetics of nucleation/crystallization during the transient cooling stage[15]. The size and size distribution of the crystallites are dramatically affected by cooling rate. With a fast cooling rate, the size would be smaller and the size distribution would be narrower, and vice versa[15]. On the other hand, during the late stage of crystallization, small individual crystallites will agglomerate to form big grains due to the coarsing process. To achieve small crystallites and hence small pores in final membranes, it is important not only to get small crystallites at the first place, but also to constrain further growth of small individual crystallites during the late stage, especially to prohibit their agglomeration. The second challenge is much more difficult to tackle, and to the best of our knowledge, it has not been worked out by other researchers.

In principle, the growth of solvent crystallites can be sterically hindered if significant enrichment of polymer solute occurs at the time of solvent crystallization, provided a directed polymer concentration gradient can be built in the polymer solution. And theoretically, this requirement can be fulfilled with a selected solvent whose melting point is only slightly lower than the room temperature. When a polymer solution film containing the solvent is unidirectionally cooled from one side to a temperature well below the freezing point of the solvent, a temperature gradient would be built in the polymer solution film. At the colder side, accompanying the nucleation/crystallization of the solvent, the remaining polymer solution would enter into unstable region in the phase diagram and start to demix due to both the loss of the solvent in the liquid phase and the reduced solubility of polymer at lower temperatures. Upon demixing/phase separation, the polymer starts to precipitate, which leads the polymer concentration in the remaining liquid phase to be much smaller than the adjacent polymer solution at higher temperatures and drives polymer solute to diffuse towards the cold end, forming a denser layer than the warmer parts. It is apparent that the amount of diffused polymer solute is determined by the diffusivity of the polymer, the polymer concentration difference, which is affected by the temperature gradient and the time available for diffusion before the liquid phase is frozen. In an ideal condition, enough polymer solute can diffuse to the cold end accompanying the nucleation/crystallization of the solvent, and fill into the space between solvent crystallites, thus sterically hindering the agglomeration of the crystallites to remain their small size.

In this study, we use a 20 wt.% PVDF solution in dimethyl sulfoxide (DMSO, melting point at 18.55 °C, a widely used green and safe solvent approved by FDA) to demonstrate the feasibility of the proposed combined crystallization and diffusion (CCD) method. With this new method, PVDF membranes with pore size down to 30 nm are prepared. The membranes have a unique and optimized structure, and show superior water permeation flux compared with traditional NIPS and TIPS membranes with similar pore sizes. The membranes also show high permeation flux after fouling and excellent mechanical properties, which are all crucial in real applications.

## Results

**Membrane formation and structures**. During the preparation process using the CCD method, the membrane structure is closely related to the cooling rate. Typically, the polymer solution was cast onto a plate with thickness of 1 mm, and then was unidirectionally cooled to a temperature well below the freezing point. The cooling rate was manipulated by contacting the casting plate with a pre-cooled cold plate ( − 30 °C) on a freezing board, or by immersing the cast polymer film into pre-cooled hexane ( − 15 °C) or liquid nitrogen ( − 196 °C). For the former cases, the material of the pre-cooled cold plate and the casting plate is aluminium or glass to realise the different thermal conductions and thus different cooling rates; for the later cases, a 1 cm thick glass casting plate was used to ensure fast cooling from only one side. After cooling, the frozen film was immersed into iced water to leach the DMSO out and final membranes were then formed.

Figure 1a shows the calculated temperature changes at the position of 10 µm from the cold-end interface of the polymer film, and Fig. 1b shows the temperature profiles within 200 µm from the cold end of the polymer film after cooling for 1 s. Figure 1c gives a cross-sectional scanning electron microscopy (SEM) image of the final membrane prepared by using a glass cold plate and a glass casting plate (denoted as Glass/Glass). The prepared membrane has an asymmetric structure with gradually open micro-channels, which have been commonly observed in freeze-drying process when unidirectional cooling is applied due to the Mullins–Sekerka instability[10,16,17]. In addition, pores smaller than 1 µm can be seen at the cold-end edge and surface (Fig. 1d,g). With a faster cooling rate realized by using a glass cold plate and an aluminium casting plate (denoted as Glass/Al), the pore size however can be quickly reduced to about 100 nm, as shown in Fig. 1e,h. And with an even faster cooling rate

realized by an aluminium cold plate and an aluminium casting plate (denoted as Al/Al), the pore size can be dramatically reduced to 30–50 nm, as shown in Fig. 1f,i. Apparently in the case of Glass/Glass and Glass/Al samples, the slow cooling rates would have produced bigger initial DMSO crystallites than the Al/Al sample, and the temperature gradients in the cold-end region are also much less steep than the latter case, which would have resulted in slower polymer diffusions. Both factors would contribute to the formation of bigger pore size in Glass/Glass and Glass/Al samples, and it is difficult to tell which factor is more important during the membrane formation process. However, those cases using pre-cooled hexane and liquid nitrogen clearly show the importance of polymer diffusion. When immersed into the hexane and liquid nitrogen, the polymer film underwent much faster cooling than the Al/Al case, and the initial DMSO crystallites in principle should be smaller than the

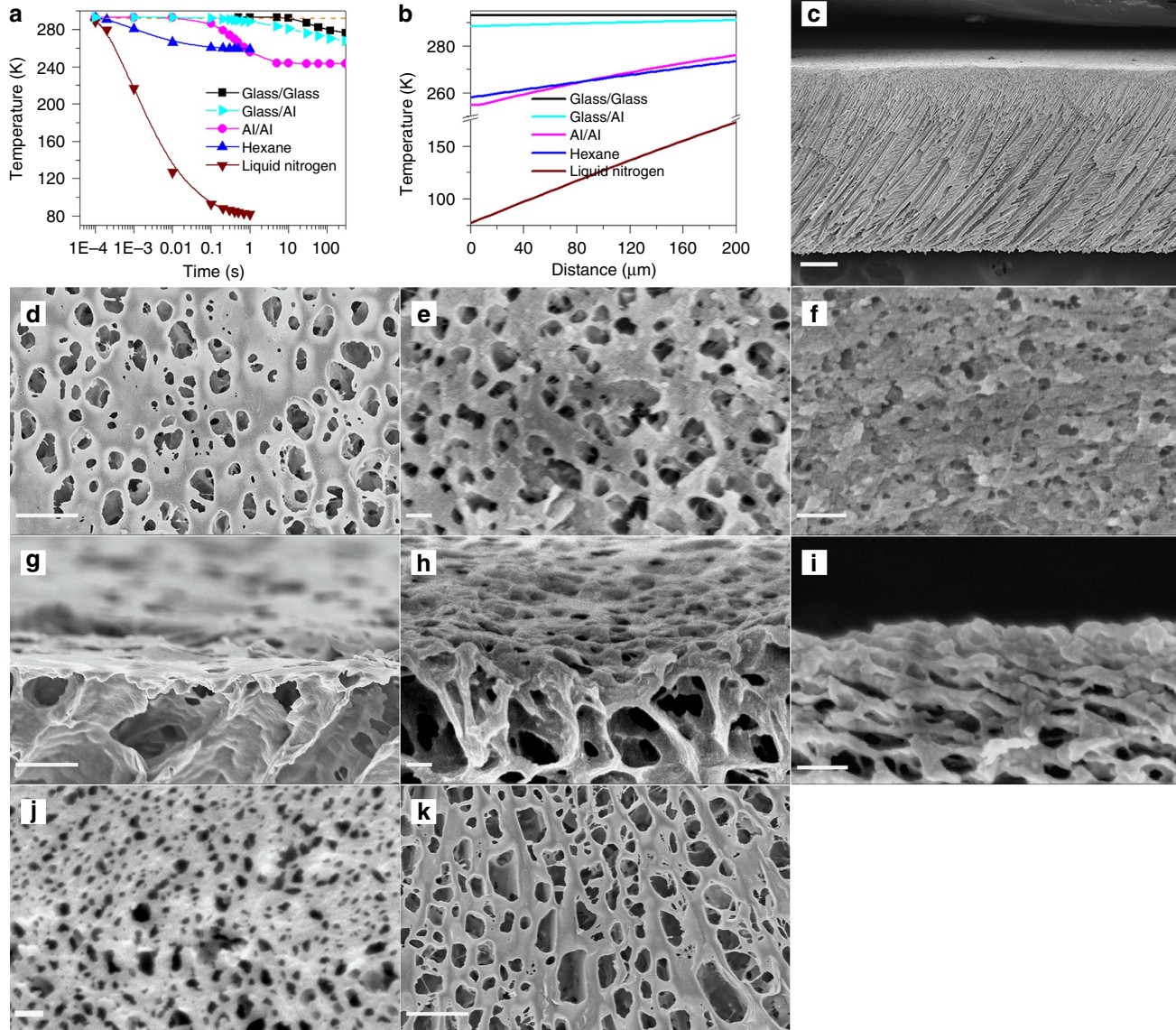

**Figure 1 | PVDF membranes prepared by the combined crystallization and diffusion method with different cooling rates.** (**a**) Temperature change in the polymer film at the position 10 µm away from the cooling interface; the left of the slash is the material of the cold plate, and the right is the material of the casting plate. The brown dashed line shows the melting point of the solvent DMSO. (**b**) Temperature profile of the polymer film from the cold end after cooling for 1 s. (**c**) Cross-sectional SEM image of the Glass/Glass sample. (**d**–**f**) Cold-end surface and (**g**–**i**) top layer cross-sectional SEM images of the Glass/Glass, Glass/Al, Al/Al sample, respectively. (**j**) Hexane and (**k**) liquid nitrogen sample's surface SEM image. The scale bar in **c** is 100 µm; in **d,g,j** and **k** is 2 µm; and in **e,f,h** and **i** is 200 nm.

Al/Al case. However, the pore sizes in both cases are even bigger than the Glass/Glass case, as shown in Fig. 1j,k. In both the cases, although a temperature gradient similar to or larger than the Al/Al case was applied, the polymer film at the cold end (10 μm from the cooling interface, for example) was cooled down to less than 7 °C within 0.001 s and the polymer film would be frozen almost instantly, leaving virtually no time for polymer to diffuse. Taking the diffusivity of polymer solute in liquid DMSO as $1 \times 10^{-9}\, m^2\, s^{-1}$, which is a typical value for polymer diffusion in solvents, the maximum distance that the polymer can travel within 0.001 s could be calculated to be only 1.4 μm, implying there was essentially no meaningful polymer diffusion occurred. As a consequence, the initial DMSO crystallites were allowed to agglomerate to form bigger grains resulting in big pores in the final membranes.

In the above cases, the 20 wt% PVDF/DMSO solution has a high freezing point of 14.3 °C (lower than the melting point of pure DMSO due to the presence of polymer), and no phase separation of the solution was observed before freezing, which confirms early solvent crystallization before the phase separation of the polymer solution. On the other hand, if the sequence of phase separation and solvent crystallization is altered, the results will be totally different. In two other Al/Al cooling cases (−30 °C), we used N-methyl-2-pyrrolidone (NMP) and dimethylacetamide (DMAc) as the solvent, whose melting points are considerably lower than DMSO (−24 °C and −20 °C, respectively), deliberately let phase separation induced by the reduction of solvent power to happen before the crystallization of the solvent. The reversed sequence of phase separation and solvent crystallization in both polymer solution was confirmed experimentally, as phase separation was observed at 5 °C for the PVDF/DMAc solution and −24 °C for the PVDF/NMP solution, whereas the freezing point was −25 °C for the former, and lower than −28 °C for the later. In both the cases, owing to the precipitated solid polymer blocks that lead steric hindrance effect to lateral solvent crystallization, the formation of micro-channels was prevented and the membranes have quite homogenous structures (Supplementary Figs 1 and 2 and Note 1), which are commonly observed in TIPS membranes. And in both the membranes, a thick, dense separation layer was formed at the cold side because of the low freezing points of the PVDF/NMP and PVDF/DMAc solutions, which gave prolonged time for PVDF solute to diffuse before solvent crystallization. Since the dense separation layer was formed before solvent crystallization, no solvent crystallites are expected to present in this layer, and in fact no visible pores were found in this layer under high-resolution SEM.

**Permeation characteristics and performance-structure rationale.** The CCD pure PVDF membranes made from DMSO solution show very narrow pore size distributions in the separation layer (Supplementary Fig. 3 and Note 2) and superior permeation performances compared with the membranes made by the conventional NIPS method. Table 1 summarizes the permeation characteristics of these CCD membranes and the NIPS PVDF membranes. For the microfiltration CCD membranes with pore sizes of 119 nm and 345 nm, the pure water flux reached stunning 5,017 and 10,998 LMH bar$^{-1}$, respectively. And for the ultra-filtration membranes, it shows that the pure water permeation fluxes of the CCD membranes are substantially higher than the NIPS membranes. The CCD Al/Al membranes showed pure water fluxes of up to 861 LMH bar$^{-1}$, which are two orders of magnitude higher than the control NIPS membranes with similar pore size. Table 1 also gives the permeation characteristics of the CCD Al/Al membranes with different casting thicknesses. A very

interesting correlation between the permeation flux and membrane thickness was found, that is, the flux increases as the thickness increases. It can be seen that the pore size did not show significant changes with the thickness, but the flux increased from 486 LMH bar$^{-1}$ for the 100 μm thick membrane gradually to 861 LMH bar$^{-1}$ for the 1 mm thick membrane. This trend can be attributed to different PVDF diffusion rates during the unidirectional cooling. It can be calculated that by changing the thickness of the cast polymer film, the cooling rate at the cold end was almost not affected during the time of interest (Supplementary Fig. 4a). However, the temperature gradient increases when the thickness reduces (Supplementary Fig. 4b), which would provide a larger driving force for the polymer solute to diffuse to the cold end and thus form a denser and thicker separation layer, leading to a smaller permeation flux. The changes in the thickness and the density of the separation layer are clearly revealed by SEM images (Supplementary Fig. 4), which agree very well with the trends of the pure water flux and the temperature gradient. As a comparison, such a trend has not been observed in PVDF membranes also using DMSO as the solvent but via the conventional NIPS method. The permeation flux of the CCD membranes can be further enhanced by improving the hydrophilicity of the pores, since pure PVDF is widely considered to be a hydrophobic material[1,2]. Table 1 also listed the permeation characteristics of two typical modified CCD PVDF microfiltration and ultrafiltration membranes, which were improved simply by blending a hydrophilic polymer polyethylene glycol (PEG) with PVDF. The improvement is especially obvious for the Al/Al ultrafiltration membrane whose pure water permeation flux increased from 500-600 to about 1,400 LMH bar$^{-1}$ after modification, while the pore size was kept unchanged. Both the modified and unmodified CCD membranes showed significantly higher fluxes than commercial PVDF membranes (which normally use modified PVDF) with similar pore size, as depicted in Table 1, suggesting that the CCD PVDF membranes have great potential to replace existing commercial membranes.

To explore why the CCD method brings such high permeability to the PVDF membranes, we have compared the structural features between the CCD and NIPS pure PVDF membranes. Figure 2 shows clearly the structural features of a CCD Al/Al PVDF membrane prepared with 0.3 mm casting thickness. In this membrane, a thin separation layer is supported by numerous very well-arranged micro-channels whose size gradually increases from the separation layer (Fig. 2a). The cross-section image of the membrane shows clearly a number of tortuous pores in the separation layer, and intensively scattered pores on membrane surface (Fig. 2b,c). Furthermore, the supporting layer of the CCD membranes is composed of fully opened, oriented and inter-connected micro-channels, which actually give negligible resistances to water permeation (Fig. 2d,e). By comparison, the NIPS membranes show typical asymmetric structures with a skinned top layer supported by a region of finger-like voids and then a sponge-like layer (Supplementary Figs 5 and 6). Although the skinned top layer of the NIPS membranes is thinner compared with the CCD membranes, only few pores on the membrane surface can be observed within the scanned area under SEM (Supplementary Figs 5 and 6), implying a very low surface porosity. On the other hand, the CCD membrane has a very porous separation layer. Besides, the NIPS membrane has a largely closed backside and back surface (Supplementary Figs 5 and 6), which would not only contribute to the total transport resistance, but also tends to intensify fouling problem because foulant would accumulate in the supporting layer in real applications. Such foulant accumulation in the supporting layer, however, can be avoided in the CCD

**Table 1 | Permeation characteristics of CCD PVDF membranes, NIPS PVDF membranes and some commercial PVDF membranes.**

| Membrane material | Membrane type | Pure water flux (LMH bar$^{-1}$) | Pore size (nm) |
|---|---|---|---|
| Pure PVDF | | | |
| | CCD membranes* | | |
| | Glass/Glass 1.0 mm | 10,998 ± 407 | 345 ± 26 |
| | Glass/Al 1.0 mm | 5,017 ± 547 | 119 ± 10 |
| | Al/Al 1.0 mm | 861 ± 78 | 45 ± 3 |
| | Al/Al 0.5 mm | 570 ± 37 | 29 ± 3 |
| | Al/Al 0.3 mm | 608 ± 82 | 30 ± 9 |
| | Al/Al 0.1 mm | 486 ± 28 | 38 ± 11 |
| | NIPS membranes* | | |
| | DMSO 1.0 mm | 6.9 ± 3.4 | 35 ± 7 |
| | DMSO 0.5 mm | 6.1 ± 1.2 | 45 ± 9 |
| | DMSO 0.3 mm | 9.3 ± 5.8 | 54 ± 12 |
| | NMP 0.3 mm | 2.3 ± 2.6 | 61 ± 11 |
| | DMAc 0.3 mm | 2.7 ± 0.5 | <18 |
| Modified PVDF | | | |
| | PVDF-PEG CCD membrane* | | |
| | Glass/Al 1.0 mm | 6,649 ± 675 | 162 ± 1 |
| | Al/Al 0.3 mm | 1,384 ± 112 | 38 ± 2 |
| | Commercial membranes† | | |
| | DOW | 40–120 | 30 |
| | QUA | 20 | 40 |
| | KOCH PURON | 100 | 30 |
| | GE ZeeWeed 1500 | 135 | 20 |
| | TORAY | 30 (MBR conditions) | 80 |
| | Pall | >3,000 | 200 |
| | Pall | >8,200 | 450 |
| | TriSep TM10 | 90 | 200 |
| | Hydranautics HYDRAcap | 34–110 | 80 |

*Sample names are ended with casting thickness; pore sizes were determined by the gas–liquid displacement method.
†Pore sizes are nominal pore sizes provided by the manufacturer; water fluxes were converted from product brochure of membrane modules, but operation pressures and other conditions are unclear.

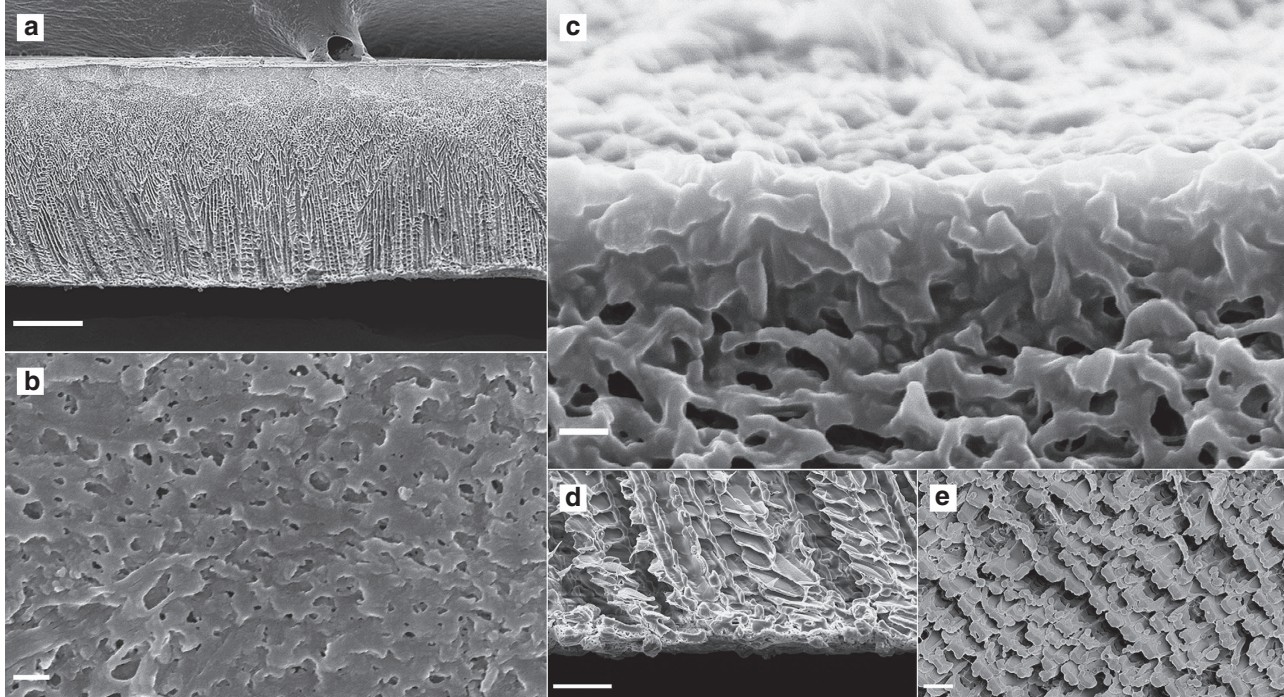

**Figure 2 | SEM images of the CCD Al/Al PVDF membrane prepared with a 0.3 mm casting thickness.** (**a**) Cross-sectional overview; (**b**) pores on the surface; (**c**) pore structure in the separation layer; (**d**) cross-sectional view of inter-connected micro-channels at the back side and (**e**) opened micro-channels on the back surface. The scale bar in **a** is 50 µm; in **b** and **c** is 200 nm; and in **d** and **e** is 10 µm.

membranes. Fouling tests with a 1 g l$^{-1}$ bovine serum albumin (BSA) solution reveal that the CCD Al/Al 1.0 mm pure PVDF membranes have lower tendency of fouling, which showed a slow and gradual decline of the permeation flux from 300 LMH to a steady flux of 100 LMH after being tested for 24 h (Supplementary Fig. 7a). Considering that the tendency of fouling

is closely related to permeation flux[18], that is, higher permeation fluxes normally lead to more severe fouling, the fouling rate of the CCD membrane is impressive compared with other NIPS and TIPS pure PVDF membranes[19–22]. As being recommended by most commercial membrane suppliers that ultrafiltration/microfiltration membrane modules should be periodically cleaned with an interval of 30–40 min after operation to resume permeability, the flux recovery of membrane after cleaning was also tested (Supplementary Fig. 7b). The pure water flux of the CCD membrane could not be fully recovered after fouling, which is in agreement with other NIPS or TIPS PVDF membranes[19–22]. However, even after fouling, the CCD ultrafiltration membranes still gave a permeation flux of ~ 200 LMH bar$^{-1}$, which is about one order of magnitude higher than conventional PVDF membranes (Supplementary Table 1). The CCD fabrication process can bring optimized membrane structures, but cannot alter the nature of the membrane material. It has been well known that pure PVDF material has high affinity to proteins, therefore, BSA is difficult to be removed from the membrane surface. On the other hand, the surface nature of the membrane can be changed by modification to reduce fouling, as it has been intensively studied in the membrane communities[1,2,19,22,23], and it would further improve the anti-fouling property.

**Mechanical properties.** Fourier transform infrared spectroscopy (FT-IR), X-ray diffraction and differential scanning calorimetry (DSC) results show that the CCD pure PVDF membranes are composed of mainly β-phase and also γ-phase PVDF crystallites with a crystallinity of around 60% (Supplementary Figs 8–10). The tightly connected PVDF grains and grain boundaries are clearly shown under SEM, especially in the supporting layer of the membranes (Supplementary Fig. 11). The formation of tightly connected PVDF grains and grain boundary might be attributed to solvent crystallization: after DMSO forms solid crystals, the PVDF is jammed into the space between DMSO crystals, and therefore the PVDF is compressed, forming rather dense PVDF grains. This is distinct from the NIPS membranes, in which a large portion of α-phase PVDF presents (Supplementary Figs 8 and 9), and PVDF grains are normally loosely connected (Supplementary Figs 5 and 6). Owing to the tightly connected PVDF grains presented in the supporting layer that make sliding along grain boundaries difficult, and perhaps together with the high crystallinity of PVDF and the absence of α-phase[24–26], the CCD membranes are more rigid compared with conventional NIPS membranes, which is reflected by low elongation ratios at the breaking point ( < 50% as listed in Supplementary Table 2), The elongation ratio of the CCD membranes is affected by the cooling rate, which might be related to the microstructural difference of the CCD membranes (Supplementary Note 3), whereas for commercial NIPS PVDF membranes, the elongation ratio is often higher than 100%, considerably higher than the CCD membranes. It is also worth mentioning that the membranes could potentially benefit from the major β-phase PVDF, which is a well-known piezoelectric material that may produce micro-vibration under an alternating electrical field, to realize self-cleaning functions and thus help to mitigate the notorious membrane fouling problems[27].

Together with the unique structure of very well-oriented micro-channels and gradually changed pore size, the high rigidity helps the CCD membrane to resist high pressures. As an example, the CCD Al/Al membrane with 1 mm casting thickness tested under a high pressure at 34.5 bar was able to maintain their thickness (Supplementary Fig. 12). On the contrary, the NIPS membranes were severely compressed after the same test and the thickness was reduced to three-fourth the original value

(Supplementary Fig. 12). The CCD membranes can even withstand high-pressure mercury intrusion porosimetry tests that give information on both the membrane supporting and separation layers, whereby NIPS membranes cannot handle. The mercury intrusion results of the CCD membranes show an overall porosity of about 75–76% and a broaden pore size distribution from around 20 μm to less than 0.1 μm, which reflects the gradual pore size change from the supporting layer to the separation layer (Supplementary Fig. 13 and Note 4). The CCD membranes have also shown excellent resistance to abrasion, which is commonly found in practice that shortens the membrane lifetime. After an accelerated abrasion test, the CCD Al/Al membrane could maintain its original pore structure, whereas NIPS membranes were severely damaged (Supplementary Figs 14–17 and Note 5). Furthermore, since the CCD membranes show increased fluxes when the thickness increases, excellent mechanical properties and high fluxes can be obtained simultaneously. For example, the CCD Al/Al membranes with 1 mm casting thickness and 1 cm width showed a 12 Newton fracture tensile force in the tensile test (Supplementary Table 2). This means that such a flat-sheet membrane of dimensions $1 \times 2 \, m^2$ (width × length) can handle the drag force produced by flowing water along the length direction with a speed of 23.6 m s$^{-1}$, which is much higher than the practical flow speeds used in real applications (normally less than 6 m s$^{-1}$).

## Discussion

In summary, we have proposed a new procedure for nano-structured membrane manufacturing with the effective pore size down to 30 nm achieved in this study. The CCD membranes have shown excellent permeation performance and mechanical properties overwhelming traditional NIPS membranes, and they are of great potential to upgrade existing filtration units. The manufacturing process based on the proposed mechanism is of much less influencing factors compared with conventional standard NIPS approach and thus is highly reliable with reproducible results (see the comparison in Supplementary Table 3). The principles can also be easily adapted to other commonly used membrane materials such as polyethersulfone and cellulose acetate, and it is, thus, expected to open up a new route for manufacturing high-performance membranes with nanostructured pores using different membrane materials.

## Methods

**Materials.** Commercial polyvinylidene fluoride (PVDF, Kynar K-761, $M_w = 440,000$ Da, $\rho = 1.79 \, g \, cm^{-3}$) was purchased from Elf Atochem and was dried at 80 °C for 24 h before use. Polyethylene glycol (PEG-400, $M_n = 400$), bovine serum albumin (BSA), dimethyl sulfoxide (DMSO), dimethylacetamide (DMAc), N-methyl-2-pyrrolidone (NMP), ethanol, hexane, SiC, were purchased from Sigma-Aldrich, UK and were used as received.

**Membrane preparation.** Pure PVDF flat sheet membranes with highly asymmetric and self-assembled ordered structure were produced by a CCD method. The PVDF dope solutions were prepared by dissolving PVDF powder (20 wt.%) in DMSO at 80 °C or in NMP and DMAc at room temperature, and then was left in the oven at 80 °C overnight to remove bubbles. The dope solution was then casted on a casting plate of certain thickness and was then unidirectionally cooled to a pre-determined temperature in two ways. As shown in Supplementary Fig. 18, one was transferring the casting plate onto a pre-cooled cold plate ( − 30 °C, detected by an infrared thermometer) on a freezing board, and the other was immersing the casting plate into a pre-cooled liquid such as hexane ( − 15 °C) or liquid nitrogen ( − 196 °C). For the former cases, the material of the cold and casting plates was aluminium or glass to realize different thermal conductions and thus different cooling rates; for the later cases, a 1 cm thick glass casting plate was used to ensure fast cooling from only one side. After cooling, the frozen casting film was immersed in iced water to leach the solvent out. The water was changed regularly to remove the residual solvent. Apart from the materials of the plates, the casting film thickness was varied to investigate its effects on the membrane morphology and properties.

Modified PVDF membranes were also prepared with the same method, except the polymer solution is blended with PEG-400 with the PEG:PVDF:DMSO mass ratio of 1:4:16.

Besides, conventional non-solvent induced phase separation (NIPS) method was used to prepare PVDF membranes as the control samples using DMSO, DMAc and NMP as the solvent and deionized water as the non-solvent. The polymer solution was cast on a glass plate at room temperature and then immediately immersed into water bath. The fabricated membrane was then kept in deionized water, which was changed frequently to remove the residual solvent before all the characterizations.

The preparation conditions of each sample were summarized in Supplementary Table 4.

**Membrane characterization.** The wet membranes were used directly for filtration tests, gas–liquid displacement porosimetry and the abrasion test, but were dried via solvent (ethanol) exchange technique before other characterizations.

**Scanning electron microscopy.** The morphologies of the membrane samples including separation layer, supporting layer and cross section were observed by SEM (LEO Gemini 1525 FEGSEM, Tokyo, Japan). The wet membranes were first immersed in ethanol for 30 min to replace all the water inside the pores, and then were fractured in liquid nitrogen to obtain the cross-sectional samples. The prepared samples were coated with gold of 10 nm thickness before SEM observation.

**FTIR spectroscopy.** The phase structure of PVDF membranes was analysed by using a FT-IR spectrometer (PerkinElmer, Spectrum One equipped with an attenuated total reflection attachment). The samples were placed on the sample holder and all the spectra were recorded in the wavenumber range of $4{,}000{-}600\,\mathrm{cm}^{-1}$ by accumulating eight scans at a resolution of $2\,\mathrm{cm}^{-1}$.

**Differential scanning calorimetry.** The melting behaviour of each membrane sample was characterized by DSC (Pyris-1, PerkinElmer, Beaconsfield, UK) and was used to determine the percentage crystallinity of PVDF in the membranes. The samples were heated from 20 °C to 220 °C at 10 °C min$^{-1}$. The percentage crystallinity of PVDF in each membrane sample was calculated by the equation shown below:

$$\% \ \mathrm{Crystallinity} = \frac{\Delta Hm}{\Delta Hm^{o}} \times 100\%$$ (1)

where $\Delta Hm$ is the heat associated with melting (fusion) of the membrane and is obtained from the DSC thermograms, $\Delta Hm^{o}$ is the heat of melting if the polymer was 100% crystalline and is $104.7\,\mathrm{J\,g}^{-1}$ for PVDF.

**X-Ray diffraction analysis.** The crystalline structure of all the membrane samples was determined using an X-ray diffractometer (X'Pert PRO Diffractometer, PANalytical) with a voltage of 40 kV and current of 40 mA. All the samples were characterized in the scanning range of $5° < 2\theta < 50°$.

**Pure water permeation test.** To evaluate the membrane permeability, pure water permeation tests were conducted using a 300 ml dead-end filtration cell (HP4750 Stirred Cell, Sterlitech Corporation, USA). The PVDF membrane samples prepared by the CCD method were tested directly at 1 bar without any pre-treatment such as membrane compaction at higher pressure. This is because the membrane samples prepared by such method possessed excellent mechanical strength and could withstand at high pressure without any flux decline being observed. On the other hand, the membrane samples prepared by the conventional NIPS method were compacted at a pressure of 2 bar for 30 min before sample collection at 1 bar. The permeance of the membrane was calculated on the basis of the equation shown below:

$$J = \frac{V}{A \times t}$$ (2)

where $J$ is the water flux, $V$ is the permeate volume, $A$ is the effective membrane area, $t$ is the time of permeate collection.

**BSA fouling test.** The fouling test was conducted using a cross-flow filtration cell (CF042 Crossflow Assembly, Sterlitech Corporation, USA) and BSA was used as a model protein to investigate the fouling resistance of the membrane samples. In the test, $1.0\,\mathrm{g\,l}^{-1}$ BSA aqueous solution was circulated through the feed side of the filtration cell, and the weight of permeate was recorded by a computer in real time. For permeability recovery tests, in each cycle, pure water was used as the feed first, and then followed by feeding BSA solution. The membrane after BSA fouling was cleaned in ultrasonication bath for 5 min and repeated three times before the next cycle. In each test, a constant 1 bar transmembrane pressure was applied.

**Gas–liquid displacement porosimetry.** In this work, the membrane pore size and pore size distribution were characterized by the gas–liquid displacement method using POROLUX 1000 (POROMETER nv, Belgium). The wet membrane was cut into certain size and wetted with a specific wetting liquid, POREFIL (POROMETER nv, Belgium, surface tension of $16\,\mathrm{mN\,m}^{-1}$). In the test, the pressure of the testing gas $N_2$ was increased from 0 to 34.5 bar step by step to replace the wetting liquid inside the membrane pores. At each step, both the pressure and the flow had to be stabilized within $\pm 1\%$ for 2 s before the data were recorded. The relevant pore size corresponding to each operating pressure can be calculated on the basis of the Young–Laplace equation:

$$d = \frac{4\gamma \cos\theta}{\Delta P}$$ (3)

where $d$ is the diameter of the pores behaving as gas paths and contributing to the gas flow at each operating pressure; $\gamma$ is the surface tension of the wetting liquid, which is $16\,\mathrm{mN\,m}^{-1}$; $\theta$ is the contact angle of the wetting liquid on the membrane surface, which is 0°; $\Delta P$ is the specific operating pressure.

Only the neck size of open pores could be measured using this method and for each sample, the mean flow pore diameter and pore size flow distribution were obtained.

**Mercury intrusion porosimetry.** For two typical CCD membrane samples, Glass/Al 1.0mm and Al/Al 1.0mm, mercury intrusion data were also collected at absolute pressure ranging between $1.38 \times 10^{3}$ and $2.28 \times 10^{8}\,\mathrm{Pa}$ (0.2–33,500 p.s.i.a.) (Micromeritics Autopore IV) with an equilibration time of 10 s and assuming a mercury contact angle of 130°. The flat sheet membranes were cut into sections of approximately 4 mm in diameter before mercury intrusion analysis.

**Mechanical test.** Mechanical properties of the membranes were tested according to American Society for Testing and Materials (ASTM) D882 using a tensile testing machine, Lloyd EZ 50. The samples were cut into 10 mm wide parts and the thickness was measured with micrometre. Each sample was initially fixed at a gauge length of 50 mm and was then stretched at a constant rate of 10 mm min$^{-1}$; the corresponding tensile force was recorded by a transducer. The elongation ratio and tensile strength at the breaking point and Young's modulus were measured. At least five samples were tested for each membrane and the averaged value was recorded.

**Abrasion test.** In this work, the abrasion test was carried out using a 400 ml dead-end filtration cell (Stirred Cell Model 8400, Merck Millipore, Germany). A $2{,}000\,\mathrm{mg\,l}^{-1}$ silicon carbide suspension was prepared and used to simulate the accelerated abrasion condition in wastewater treatment. The wet membrane sample was placed in the filtration cell, and 300 ml of the SiC suspension was filled and then stirred at 400 r.p.m. for 2 weeks. Subsequently, the membrane sample was washed under ultrasound for 10 min to remove all the debris worn away from the membrane during the test. Then the change in the membrane structure was observed using SEM. It is known from previous work that the most severe damages occur at the centre part of the membrane[28]; therefore, all SEM images given here were taken from the centre of the membranes for fair comparison.

**Determination of phase separation and freezing temperatures of PVDF solutions.** About 5 ml PVDF solutions (20 wt.%) using DMSO, DMAc or NMP as solvent were sealed in 10 ml glass vials and then gradually cooled in a hexane bath. For the temperature range from 25 to 3 °C, the temperature was step changed with a chiller and dwell at each set temperature for 3 min; for the temperature range from 3 to −28 °C, the temperature was reduced slowly but continuously, therefore the determined temperatures have an error within 1 °C. Phase separation was revealed by laser scattering, and the freezing of polymer solution can be noticed visually.

**Calculation of temperature profile and cooling rate.** Transient temperature profiles across casting polymer films and the cooling rates at fixed positions in the casting film were calculated by the commonly used finite difference method with the explicit scheme[29]. The one-dimensional heat conduction models for the scenarios involved in this research were set as below.

**Cooling with a pre-cooled cold plate.** The setting of the initial conditions for the calculation of thermal conduction and temperature profile under the circumstance of cooling with a pre-cooled cold plate is illustrated in Supplementary Fig. 19.

The boundary of the cold plate was fixed at −30 °C, as it was continuously cooled by a freezer. A 20 mm air gap was used to allow the temperature of the top surface of the polymer casting film to change, and the boundary was fixed at 20 °C. The thickness of the air gap has a little influence on the final temperature of the top surface of the polymer film, but basically produces no difference within the time of interest.

Heat conduction in the layers of different materials was deemed as heat diffusion along one-dimensional grids, on which points with an interval ($\Delta x$) of 5 μm were used to solve the heat conduction equation numerically:

$$\frac{\partial T}{\partial t} = \kappa \frac{\partial^2 T}{\partial x^2}$$ (4)

where $T$ is the temperature, $t$ is the time, $x$ is the distance and $\kappa$ is the thermal diffusivity. The thermal diffusivity $\kappa$ is defined as:

$$\kappa = \frac{k}{\rho c_p} \tag{5}$$

where $\rho$ is density, $c_p$ is heat capacity and $k$ is thermal conductivity.

Inside each homogeneous material layer, heat conduction is calculated by:

$$T_i^{n+1} = T_i^n + \kappa \Delta t \left( \frac{T_{i+1}^n - 2T_i^n + T_{i-1}^n}{(\Delta x)^2} \right) \tag{6}$$

and at the interface between different material layers, heat conduction is calculated by:

$$T_i^{n+1} = T_i^n + \frac{\kappa_{i-1}\Delta t}{(\Delta x)^2}T_{i-1}^n - \left(\frac{\kappa_{i-1}\Delta t}{(\Delta x)^2} + \frac{\kappa_i\Delta t}{(\Delta x)^2}\right)T_i^n + \frac{\kappa_i\Delta t}{(\Delta x)^2}T_{i+1}^n \tag{7}$$

Here $i$ is the position on the grid, and $n$ is the number of the time step $\Delta t$, which is set to satisfy the condition $2\kappa\Delta t \leq (\Delta x)^2$ to meet the criteria of stability for the calculation[29].

**Cooling by immersing into liquid nitrogen or pre-cooled hexane.** In these two cases, same algorithm was used, but the grids and boundary conditions were different, as shown in Supplementary Fig. 20.

To simplify the calculation and to keep the calculating time within a manageable duration, and also owing to the lack of available literatures, it is assumed that the thermal diffusivity in each layer is constant within the temperature range of interest, and it does not change in the casting film even if the film is turned from liquid to solid. This will, of course, lead to some inaccuracy but will not alter the trends shown in the results. That is because within such a relatively narrow temperature range from $20\,^\circ\text{C}$ to $-30\,^\circ\text{C}$, the change in the thermal diffusivity is usually very small (except the case of liquid nitrogen, but for this case the lower temperature from $-30\,^\circ\text{C}$ to $-196\,^\circ\text{C}$ is no longer interested). And for common solvents and polymer, the thermal diffusivity usually does not change significantly when phase transformation happens. The thickness change of the casting film during the cooling process was also not taken into account, since the change was small and should not lead significant effects to the temperature profiles.

**Data availability.** The data that support the findings of this study are available from the corresponding author upon request.

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

## Acknowledgements

The authors gratefully acknowledge the research funding provided by EPSRC in the United Kingdom (Grant no EP/J014974/1) and B. Wang gratefully acknowledges the Marie Curie International Incoming Fellowships (Grant no. 627591).

## Author contributions

All the authors conceived the research, J.J. and B.W. did the experiments, B.W. and J.J. analysed the results and drafted the manuscript. All the author revised and proof read the paper.

## Additional information

**Competing financial interests:** The authors declare no competing financial interests.

**How to cite this article**: Wang, B. *et al.* Crystal nuclei templated nanostructured membranes prepared by solvent crystallization and polymer migration. *Nat. Commun.* 7:12804 doi: 10.1038/ncomms12804 (2016).

