## [Peer review file · Nature Communications]

Reviewers' comments:

Reviewer #1 (Remarks to the Author):

Li and co-authors reported the fabrication of nanoporous membranes by a 'freezing' approach. Asymmetric porous membranes/structures can be easily obtained by a freezing approach but usually the pore sizes are in the micron or large nanosized region. Thus it is an exciting progress to prepare the membranes with the pore sizes down to 30-50 nm. Such membranes also show much higher water permeation flux than the membranes produced by conventional phase separation method, combined with high mechanical stability and antifouling properties. The materials have been characterized to give evidence and elucidate the membrane's superior performance.

The authors have proposed the solvent crystallisation and polymer diffusion as the formation mechanism. More evidence is required although it sounds reasonable. It is widely accepted that the polymer molecules can be pushed away from the ice frozen and thus concentrate the solutions near the freezing front. This is opposite to the explaining the authors have given. Also, based on the authors' reasoning, the solvent small crystals should still suspend in the solution so that the polymer can diffuse around the crystals and prevent further growth. Can such crystals be detected to show the similar sizes as the pore sizes? Are there any other literatures to support the authors' explanation?

For the method 2, how was the cold plate immersed in liquid nitrogen? During the rapid freezing, wouldn't the vaporizing N₂ disturb the liquid film structure? How was the thickness of the liquid film controlled? For such thin liquid film, when moved closer to cold liquid, particularly liquid nitrogen, could part of the liquid film start to freeze in an uncontrolled way?

As shown by the authors with different solutions, freezing rate and polymer diffusion need to be finely balanced to produce such nanoporous membranes. Can the authors outline the selection criteria so that this method may be successfully applied to other systems?

For the NMP and DMAc solutions, evidence for the phase separation occurring prior to crystallisation?

I am not sure if Hg intrusion porosimetry can be used to show higher compression stability. Due to the surface tension of Hg, Hg may be easily compressed into pores of different size at different compressing pressure. Membranes with closed pores may not hold to the pressure. This means the CCD membranes may be highly interconnected porous so that they could withstand high-pressure Hg intrusion.

Reviewer #2 (Remarks to the Author):

In this work, a new concept of membrane manufacturing called combined crystallization and diffusion (CCD) method was proposed. The paper was mainly focus on the comparison with CCD membranes and conventional NIPS membranes in which the CCD membranes showed excellent performances. However, before publication some suggestions as follows:

1. Page 5, "Upon demixing, the polymer starts to precipitate instantly, which leads the polymer concentration in the remaining liquid phase to be much smaller than the adjacent polymer solution at higher temperatures and drives polymer solute to diffuse towards the cold end, forming a denser layer than the warmer parts. It is apparent that the amount of diffused polymer solute is determined by the diffusivity of the polymer, the polymer concentration difference, which is affected by the temperature gradient and the time available for diffusion before the liquid phase is frozen." However, in the Figure S14 of Supplementary Materials, the resultant membranes showed finger-like structure, the authors claimed that "The image shows the cross section view close to

the back side, which clearly depicted tightly connected PVDF grains and the grain boundary". Please give detail information on the "clearly depicted tightly connected PVDF grains and the grain boundary". Is it connected with the mechanical properties? In my opinion, the finger-like structure always leads to poor mechanical properties.

2. Page 6, line 10: "the polymer film at the cold end was cooled down to less than 7 {degree sign}C within 0.001 s and the polymer film would be frozen almost instantly." How to control "less than 7 {degree sign}C within 0.001 s"?

3. Page 8, line 2: In my opinion, membrane surface hydrophilicity is more important than pores.

4. Page 9, line 16: The CCD membranes are more rigid compared with conventional NIPS membranes. Why elongation ratios showed worse values than commercial NIPS PVDF membranes?

5. Table S3 of Supplementary Materials showed the mechanical properties of resultant PVDF membranes fabricated via CCD method, the differences between the mechanical properties of Al/Al 1.0 mm, Glass/Al 1.0 mm and Glass/Glass 1.0 mm were not obvious, please explain the reason.

6. Page 9, last line and Supplementary Fig. S15: The authors should show more clearly the thickness of membranes in Fig.S15(A-E).

7. Page 11, line 10: Why were the PVDF dope solutions prepared by dissolving PVDF powder in DMSO at 80 {degree sign}C while in NMP and DMAc at room temperature? Why did it make in the same condition?

8. Page 11, line 15: How to operate in such a low temperature and how to make sure the temperature was consistent during the process?

9. Is the flux recovery rate of PVDF membranes prepared via CCD method higher than that of via NIPS or TIPS after BSA fouling was tested by cleaning?

10. Some researches have demonstrated that β phase of PVDF with polarity is easy to adsorb protein. What was the absorption performance of PVDF membrane prepared by CCD method for BSA?

11. In the process of membrane fabricating, the added PEG-400 would result in crystallizing. How does PEG-400 affect the structure of membranes?

12. Was the gel temperature of the casting solution?

Reviewer #3 (Remarks to the Author):

Li's group developed a new method of PVDF UF/MF membrane fabrication. In the proposed method precipitation of solvent liquid and diffusion of solute polymer are controlled by changing the freezing point of the solvent and the unidirectional temperature gradient. They have achieved significantly high pure water fluxes compared to the membranes of equal pore sizes fabricated by NIPS or TIPS. They have also shown the benefit of their membranes such as higher mechanical strength and better antifouling capacity. The rational behind the novel membrane preparation is clearly demonstrated and well supported by the experimental data.

From the practical application of UF/MF operation, the flux becomes almost the same regardless of the significant difference in pure water flux in the presence of macromolecular solutes such as proteins. Even though they have shown that the rate of flux decline for BSA feed is much less for their CCD membrane than NIPS and TIPS membrane, comparison was not made between the fluxes at the steady state. As well, the flux recovery by membrane cleaning has not been shown. The latter information is extremely important to know the degree of reversible and irreversible fouling.

Fouling is likely affected by the pore size distribution. Table 1 does not include the pore size distribution, the measurement of which is possible by the gas-liquid method they adopted. I would like to suggest publication if they can address my comments properly.

Response to the Referees

This file lists responses to reviewers' comments point by point. We would like to use this opportunity to express our gratitude towards the reviewers who took their time to review this submission and gave constructive advice.

Reviewer #1 (Remarks to the Author):

Li and co-authors reported the fabrication of nanoporous membranes by a 'freezing' approach. Asymmetric porous membranes/structures can be easily obtained by a freezing approach but usually the pore sizes are in the micron or large nanosized region. Thus it is an exciting progress to prepare the membranes with the pore sizes down to 30-50 nm. Such membranes also show much higher water permeation reflux than the membranes produced by conventional phase separation method, combined with high mechanical stability and antifouling properties. The materials have been characterized to give evidence and elucidate the membrane's superior performance.

The authors have proposed the solvent crystallisation and polymer diffusion as the formation mechanism. More evidence is required although it sounds reasonable. It is widely accepted that the polymer molecules can be pushed away from the ice frozen and thus concentrate the solutions near the freezing front. This is opposite to the explaining the authors have given.

Response: we totally agree that the concentration of polymer near the freezing front will increase, and we have point it out in the manuscript. But it is not contradictory with our explanation. The concentration of polymer in the liquid phase will increase initially when the solvent crystallise, but when the concentration is high enough to cross the spinodal line in the phase diagram (the reduction of solvent power at lower temperatures also contributes to approach the spinodal line), the homogenous polymer solution near the freezing front will enter into unstable region, and then the polymer solution will be separated to two phases: one phase is the precipitated polymer that can be considered to be solid; and the other phase is the remaining liquid phase that have much lower polymer concentration than the adjacent unseparated solution, and thus a polymer concentration gradient is built up in the liquid phase for polymer diffusion.

Also, based on the authors' reasoning, the solvent small crystals should still suspend in the solution so that the polymer can diffuse around the crystals and prevent further growth. Can such crystals be detected to show the similar sizes as the pore sizes? Are there any other literatures to support the authors' explanation?

Response: unfortunately current techniques are extremely difficult to capture the solvent crystals in the formed frozen membranes. Electronic microscopes are not suitable for this occasion, because the vacuum requirement during sample preparation and observation will remove the solvent. What we can possibly use is AFM and we have made quite a lot effort on it, but these efforts have failed due to the difficulty of maintaining low temperature and preventing moisture adsorption during observation.

Meanwhile, we do not think the solvent crystals are really "suspended" in these cases, meaning that the crystals are not completely isolated by the polymer, otherwise the formed pores will be dead

pores that prevent water to pass through. Instead, from SEM images we can see these pores are connected. What we believe is that the solvent crystals in the separation layer are connected roughly along the thickness direction, but isolated at the lateral dimensions due to polymer diffusion. This is understandable because the temperature gradient and polymer concentration gradient are all along the thickness direction, and therefore the growth of solvent crystals and the diffusion of polymer are also favoured along this direction. It can be illustrated in below diagramme:

It is widely accepted that the solvent crystals serve as pore templates in freeze drying methods. The principle of this research is similar to freeze drying except we managed to control the size of the crystals. The other possible mechanism of pore formation is by phase separation rather than the crystal templates, as in those NIPS or TIPS membranes, but the SEM images of the separation layer shows clearly that the pore shapes (which are clearly shown in Figure 2(c)) are distinct from the pores from NIPS and TIPS membranes, but concurrent with the shapes of crystals, therefore we believe it is reasonable to attribute the pore formation to the crystals.

For the method 2, how was the cold plate immersed in liquid nitrogen? During the rapid freezing, wouldn't the vaporizing N2 disturb the liquid film structure? How was the thickness of the liquid film controlled? For such thin liquid film, when moved closer to cold liquid, particularly liquid nitrogen, could part of the liquid film start to freeze in a uncontrolled way?

Response: The casting plate together with polymer film was immersed into liquid nitrogen manually (with a holder) using the fastest speed we could reach. The thickness of the polymer film were controlled by a casting knife that allow adjust of thickness. We cannot completely deny the possible disturbance from vaporised nitrogen or uncontrolled freezing, and that is why we have also tried with much milder conditions using pre-cooled hexane, which produced very similar pore size as the liquid nitrogen case, hence we believe the hexane case together with the liquid nitrogen case are enough evidence to qualitatively demonstrate the importance of polymer diffusion during the membrane formation process.

As shown by the authors with different solutions, freezing rate and polymer diffusion need to be finely balanced to produce such nanoporous membranes. Can the authors outline the selection criteria so that this method may be successfully applied to other systems?

Response: this is a very important question, and also a difficult one. We can only list some rough criteria based on our own experience, but we must stress that the criteria are based on our very preliminary study in this new topic, and there are still many open questions to be answered. We wish other researchers will also explore this new topic and make reliable criteria in the future.

Based on our experience, we think some factors are crucial to this method:

1. the polymer solution needs to be stable (free of phase separation) at temperatures higher than the freezing point. If the polymer solution is metastable, then the cooling rate needs to be high enough to overcome the kinetics of phase separation before the freezing point.
2. to obtain UF range pores, in our cases, the instantaneous cooling rate at the separation layer is recommended to be roughly in the range of 50 – 150 °C/min. But we must stress that this cooling rate range is only suitable for the 20 wt% PVDF/DMSO solution, and the change in polymer, solvent and concentration would affect the result.
3. the freezing point of the polymer solution should be preferably higher than 0 °C, so that the pore structure can be maintained when using icy water to leach out the solvent, otherwise the nano-sized pores could be destroyed if the solvent crystals melt before being leached out (although larger pore structures could be reserved if polymer re-dissolving is slow). Of course other non-solvent for leaching can be used at temperatures lower than 0 °C, but in practice it is inconvenient and uneconomical.
4. we have found that the concentration of polymer is very important to the pore size and permeation flux. The 20 wt% concentration is a proper one for the PVDF/DMSO case, and it gives a good balance between pore size and flux for the UF membranes. But for PES/DMSO cases, other concentrations should be used to achieve the balance. It is beyond the scope of this manuscript, but will be revealed in our future work.

For the NMP and DMAc solutions, evidence for the phase separation occurring prior to crystallisation?

Response: we have done such experiments to verify the assumption: about 5 ml NMP and DMAc solutions contained in glass bottle were cooled down gradually in a cooling bath (Hexane) to -28 °C, and phase separation and freezing was confirmed by both eyes and laser scattering. The DMAc solution separated at ~5 °C, and froze at around -25 °C; the NMP solution separated at around -24 °C, but not froze till -28 °C, which was the lowest achievable temperature in our lab with such a manageable way. Similarly, we could determine the freezing point of the DMSO/PVDF solution to be 14.3 °C, and we didn't see phase separation prior to the freezing point. For all cases, the freezing temperature is lower than the pure solvent because of the existence of polymer solute.

We have also included these data into the main manuscript to enhance rationale of the new CCD method.

I am not sure if Hg intrusion porosimetry can be used to show higher compression stability. Due to the surface tension of Hg, Hg may be easily compressed into pores of different size at different compressing pressure. Membranes with closed pores may not hold to the pressure. This means the

CCD membranes may be highly interconnected porous so that they could withstand high-pressure Hg intrusion.

Response: the mercury intrusion measurement went to above 200 Mpa, and we certainly have no intention to indicate that the whole membrane can work under such a high pressure. The purpose of using the mercury intrusion data is to demonstrate the gradually changed pore structure and this unique pore structure is more stable than normal NIPS membranes under pressures.

We totally agree with the reviewer that the interconnected pore structure helps the membrane to withstand higher pressure intrusion, and closed pores will be easily compressed. Furthermore, we want to argue that even open pore structures will respond differently to the high pressure intrusion, depending on how the pores are interconnected. The below scheme shows two different simplified pore structures: the first one shows a gradually changed pore structure, where the pores shrink smoothly from big to small, and that represents the case of the CCD membranes; the second type shows a bumpy shape where large volumes are connected by narrow necks, which can represent most NIPS membranes. For the CCD cases, the mercury can inch into the smaller pores smoothly with slight pressure change; but for the NIPS cases, the pressure needed to enter the necks could be high enough to compress the large volumes.

Type 1. Gradually changed pore structure

Type 2. Bumpy pore structure

Nevertheless, the mercury intrusion results still demonstrate that the CCD membranes have very good resistance to pressures, although its highest working pressure cannot be directly linked to the pressure achieved during the mercury intrusion experiments. We can argue that when the mercury front intrudes into the membrane from the more open part, the rest part of the membrane with smaller pore diameters is under a high pressure, and this part of smaller pore diameters is stable enough to take the pressure without pore collapse. The stable pore structure is also proven when measuring pure water permeation, as we've never found flux decline due to membrane compaction. But such membrane compaction has always been observed when we measure NIPS membranes.

Reviewer #2 (Remarks to the Author):

In this work, a new concept of membrane manufacturing called combined crystallization and diffusion (CCD) method was proposed. The paper was mainly focus on the comparison with CCD membranes and conventional NIPS membranes in which the CCD membranes showed excellent performances. However, before publication some suggestions as follows:

1. Page 5, "Upon demixing, the polymer starts to precipitate instantly, which leads the polymer concentration in the remaining liquid phase to be much smaller than the adjacent polymer solution at higher temperatures and drives polymer solute to diffuse towards the cold end, forming a denser

layer than the warmer parts. It is apparent that the amount of diffused polymer solute is determined by the diffusivity of the polymer, the polymer concentration difference, which is affected by the temperature gradient and the time available for diffusion before the liquid phase is frozen."However, in the Figure S14 of Supplementary Materials, the resultant membranes showed finger-like structure, the authors claimed that "The image shows the cross section view close to the back side, which clearly depicted tightly connected PVDF grains and the grain boundary". Please give detail information on the "clearly depicted tightly connected PVDF grains and the grain boundary". Is it connected with the mechanical properties? In my opinion, the finger-like structure always leads to poor mechanical properties.

Response: the denser layer (separating layer) was formed at the side contacting with the casting plate, whereas the micro-channels were formed in the supporting layer (upper part in the membrane when looking at the method 1 shown in Supplementary Figure 1). The formation of the denser layer is mainly due to the solvent nuclei and polymer diffusion, and the formation of micro-channels is due to the oriented large solvent crystals grown during the later stage.

The formation of tightly connected PVDF grains and grain boundary can be partially attributed to solvent crystallisation. After DMSO forms solid crystals, the PVDF is jammed into the space between DMSO crystals, and therefore the PVDF will be compressed, forming rather dense PVDF grains. Since the PVDF can also form crystals and the crystallinity is $\sim 60\%$ in this case, clear grain boundary can also be formed. For amorphous polymer, for example PES, similar compressed wall structure can also be observed, but without the existence of grain boundary (not reported in this paper).

In our opinion, the compressed PVDF grain and the tight connection at least partially contribute to the rigid pore structure and the low elongation. In a porous structure, deformation often relies on sliding at grain boundaries. Tight grain boundaries make sliding more difficult, and therefore deformation of the structure is hampered. We have included the discussion into the revised manuscript.

For the micro-channels in the CCD membranes (seems like fingers in NIPS membranes, but we think they are different in terms of formation mechanism and structure), we agree with the reviewer that in general they will reduce the mechanical properties. But they are also important to achieve the balance between the permeability and mechanical properties. In our practice, the membranes with these micro-channels are robust enough to deal with daily filtration tasks, and they are very easy to handle without the need of particular care. Looking into the mechanical property of the CCD membranes, the properties along the lateral dimensions such as fractural stress and Young's modulus are not better than NIPS membranes, but they are compensated by the increased thickness. However for the resistance to compaction, which is the Z-axis property, it is much better than the NIPS membranes, and it is not compromised by the micro-channels. Besides the tight connection between PVDF grains as mentioned in above discussion, we think the excellent resistance to compaction can also be related to the unique structure of the micro-channels, in which the size reduce very smoothly along the Z-axis; whereas in NIPS membranes, the diameter of the fingers always has a sudden change. To fully understand the mechanical property of these new types of membranes, detailed structural analysis using simulation techniques such as Finite Element Analysis will be needed, and it will be one of our tasks in the future.

2. Page 6, line 10: "the polymer film at the cold end was cooled down to less than 7 {degree sign}C within 0.001 s and the polymer film would be frozen almost instantly." How to control "less than 7 {degree sign}C within 0.001 s"?

Response: the cooling rate of 7 °C in 0.001 s is not a pre-designed cooling rate, it is the instantaneous cooling rate calculated based on thermal conduction when hexane and liquid nitrogen were used as the cooling media (presented in Figure 1a in the main manuscript).

3. Page 8, line 2: In my opinion, membrane surface hydrophilicity is more important than pores.

Response: we have considered the factor of surface hydrophilicity and done water contact angle measurements, as listed in a table below. We've found that the water contact angles on the CCD membranes are not lower than the NIPS membranes, therefore it seems there is no evidence to say the high water flux through the CCD membranes was due to higher surface hydrophilicity. However, we understand that water contact angle is affected by many factors, for example, surface roughness, aside the nature of the material. Considering the fact that the CCD membranes have rougher surface than NIPS membranes, it perhaps not fair to use contact angles to compare the intrinsic hydrophilicity of the membrane material. And the intrinsic wetting property inside pores should be the property of interest, but unfortunately it is difficult to know. With all these considerations, we think the contact angle results would be misleading to readers, therefore we prefer not to include these data in this submission.

Membrane Type		Water contact angle (°)
CCD membranes	Glass/Glass 1.0 mm	99.5±4.8
	Glass/Al 1.0 mm	93.4±1.6
	Al/Al 1.0 mm	89.5±3.5
	Al/Al 0.5 mm	84.5±4.2
	Al/Al 0.3 mm	76.7±7.1
	Al/Al 0.1 mm	62.7±14.8
NIPS membranes	DMSO 1.0 mm	87.7±8.6
	DMSO 0.5 mm	74.9±5.6
	DMSO 0.3 mm	70.7±2.8
	NMP 0.3 mm	66.4±11.7
	DMAc 0.3 mm	65.1±3.8

4. Page 9, line 16: The CCD membranes are more rigid compared with conventional NIPS membranes. Why elongation ratios showed worse values than commercial NIPS PVDF membranes?

Response: a rigid structure normally comes with a low elongation ratio, which are the two sides of one coin. But for the question why the CCD membrane is more rigid, or why lower elongation ratios than NIPS PVDF membranes, we have no firm answer to the question yet.

After reviewing plenty of literatures, we've found although there are many studies made discussion on the mechanical property of PVDF membranes, these studies are often contradictory. There is

none that is systemic and able to separate different factors, and therefore nothing conclusive has been made. In general, the porosity, crystallinity, the type of crystal phase (α , β , γ) affects mechanical properties, but the microstructure of the membrane (finger like, spherulitic, cellular structures) also can make significant changes in the elongation ratio etc. A summary of elongation ratio of different TIPS PVDF membranes can be found in (Kim, Kim et al. 2016), which shows the complexity of influencing factors to mechanical properties.

Back to this study, as we have discussed previously, we think the microstructure of the membrane (the compressed PVDF grains and tight grain boundaries) should be one of the reason for the low elongation rate, but we cannot deny that the high crystallinity and absent of the α -phase may also contribute to the rigid structure, as suggested in some literature. After thinking through the question from the reviewer, we have revised the sentence in the main manuscript (in red) accordingly to reflect our updated thinking.

Reference

Kim, J. F., et al. (2016). "Thermally induced phase separation and electrospinning methods for emerging membrane applications: A review." *AIChE Journal* **62**(2): 461-490.

5. Table S3 of Supplementary Materials showed the mechanical properties of resultant PVDF membranes fabricated via CCD method, the differences between the mechanical properties of Al/Al 1.0 mm, Glass/Al 1.0 mm and Glass/Glass 1.0 mm were not obvious, please explain the reason.

Response: all three CCD PVDF membranes have very similar structure, porosity, crystal phase and crystallinity, which are major factors to mechanical properties, therefore it is reasonable that they have similar mechanical properties. Of course there are some subtle differences between the microstructures, and the change in microstructure has been reflected in their mechanical properties, which has been discussed in the Supplementary Note 3 in revised Supplementary Information.

6. Page 9, last line and Supplementary Fig. S15: The authors should show more clearly the thickness of membranes in Fig.S15(A-E).

Response: as suggested by the reviewer, we have added the thickness values into the figure caption.

7. Page 11, line 10: Why were the PVDF dope solutions prepared by dissolving PVDF powder in DMSO at 80 {degree sign}C while in NMP and DMAc at room temperature? Why did it make in the same condition?

Response: PVDF is more difficult to dissolve in DMSO than in NMP and DMAc at room temperature, therefore we used 80 °C to help dissolving process. After dissolving, all solutions were stored at 80 °C to remove bubbles, at the same time, the thermal history of the PVDF solutions became similar.

8. Page 11, line 15: How to operate in such a low temperature and how to make sure the temperature was consistent during the process?

Response: the temperature is determined by the freezing board (Polyscience ANTI-GRIDDLE® Flash Freeze), which has been set to -30 °C by the manufacturer. The surface temperature of the cold

plate is confirmed by an IR temperature detector every time before membrane preparation. -30 °C actually is not a very low temperature (the temperature in normal freezer is -18 or -24 °C), and the operator only need to contact the cold item for 2-3 seconds when transferring the casting plate from the cold plate to cold water. In our practice, we haven't been troubled by the temperature, normal oven gloves or even nitrile gloves are good enough to protect the operator.

9. Is the flux recovery rate of PVDF membranes prepared via CCD method higher than that of via NIPS or TIPS after BSA fouling was tested by cleaning?

Response: We've done flux recovering tests on the 1-mm CCD Al/Al PVDF membrane and included the result in the Supplementary Figure 10, but it doesn't give better recovery rate than NIPS or TIPS membranes. The CCD fabrication process can bring optimised membrane structure, but cannot alter the nature of the membrane material. It has been well known that pure PVDF material has high affinity to proteins, therefore it is reasonable that BSA is difficult to be removed from the membrane surface. Furthermore, it is well known that the tendency of fouling is closely related to permeation flux, higher permeation flux normally leads more severe fouling. However, even after fouling, the CCD UF membranes still give a permeation flux of ~ 200 LMH bar⁻¹, which is still much higher than commercial membranes. On the other hand, the surface nature of the membrane can be changed by modification to reduce fouling, as it has been intensively studied in the membrane communities.

10. Some researches have demonstrated that β phase of PVDF with polarity is easy to adsorb protein. What was the absorption performance of PVDF membrane prepared by CCD method for BSA?

Response: we've done BSA adsorption experiments on pure PVDF membranes to answer the question. In this experiment, all membranes were prepared with a 0.3 mm casting thickness, and total 43 cm² membrane area for each sample was used. The membranes were cut to about 1X1 cm² and put into 15 ml 1 g/L BSA solution for 24 hours at room temperature. Concentration difference of BSA solution before and after the adsorption test was determined by UV adsorption at 278 nm. The membranes were then rinsed and dried at 80 °C to obtain the weight of each sample.

The result is shown in below table. Both NMP and DMAc NIPS membranes showed low BSA adsorption, but the DMSO NIPS membrane showed significantly higher adsorption, although these three NIPS membranes have very similar phase composition with the majority of α-phase PVDF. For the CCD membrane, the BSA adsorption showed even higher adsorption than the DMSO NIPS membrane. It looks in these membranes, the solvent is more important than the crystal phase.

Sample	NMP-NIPS	DMAc-NIPS	DMSO-NIPS	Al/Al CCD (DMSO)
BSA Adsorption (mg/g)	1.49±0.32	0.17±0.10	12.01±0.57	16.01±0.20

One factor might affect the adsorption results significantly, which is the accessible surface area. For the CCD membranes, because the pores are highly interconnected, the accessible surface area is also high. But for the NIPS membranes, the accessible surface area is affected by the number of dead pores, which is in turn affected by the solvent used in the polymer solution. Hence it is difficult to conclude that the β phase is easier to adsorb protein. Considering this point, the BSA adsorption

data was not included into the paper to avoid misleading to readers.

11. In the process of membrane fabricating, the added PEG-400 would result in crystallizing. How does PEG-400 affect the structure of membranes?

Response: after adding PEG-400, no obvious changes can be observed, neither the microstructure, nor the crystalline structure. FT-IR and XRD results showed no sign of α -phase, the membranes are still β and γ -phase PVDF. The crystallinity of PVDF (PEG is excluded) increased to $65\pm 1\%$ for all CCD membranes, just slightly higher than the pure PVDF CCD membranes.

12. Was the gel temperature of the casting solution?

Response: the 20% PVDF/DMSO solution start to freeze (solvent crystallisation) at $14.3\text{ }^{\circ}\text{C}$, not phase separation or gelation was observed before the freezing temperature.

Reviewer #3 (Remarks to the Author):

Li's group developed a new method of PVDF UF/MF membrane fabrication. In the proposed method precipitation of solvent liquid and diffusion of solute polymer are controlled by changing the freezing point of the solvent and the unidirectional temperature gradient. They have achieved significantly high pure water fluxes compared to the membranes of equal pore sizes fabricated by NIPS or TIPS. They have also shown the benefit of their membranes such as higher mechanical strength and better antifouling capacity. The rational behind the novel membrane preparation is clearly demonstrated and well supported by the experimental data.

From the practical application of UF/MF operation, the flux becomes almost the same regardless of the significant difference in pure water flux in the presence of macromolecular solutes such as proteins. Even though they have shown that the rate of flux decline for BSA feed is much less for their CCD membrane than NIPS and TIPS membrane, comparison was not made between the fluxes at the steady state. As well, the flux recovery by membrane cleaning has not been shown. The latter information is extremely important to know the degree of reversible and irreversible fouling. Fouling is likely affected by the pore size distribution. Table 1 does not include the pore size distribution, the measurement of which is possible by the gas-liquid method they adopted. I would like to suggest publication if they can address my comments properly.

Response: we have changed Supplementary Figure 10 to show the real fluxes during the fouling test, and also compared it with the steady flux of NIPS & TIPS PVDF membranes. We have also done flux recovery test on same type of membrane, and included it into Supplementary Figure 10. The relevant text in the main manuscript was also changed accordingly (in red). It can be seen that the recovery of the CCD PVDF membrane is not better than NIPS & TIPS membranes. The CCD fabrication process can bring optimised membrane structure, but cannot alter the nature of the membrane material. It has been well known that pure PVDF material has high affinity to proteins, therefore it is reasonable that BSA is difficult to be removed from the membrane surface. Furthermore, it is well known that the tendency of fouling is closely related to permeation flux, higher permeation flux normally leads more severe fouling. However, even after fouling, the CCD UF

membranes still give a permeation flux of $\sim 200 \text{ LMH bar}^{-1}$, which is still several times higher than commercial membranes. On the other hand, the surface nature of the membrane can be changed by modification to reduce fouling, as it has been intensively studied in the membrane society.

The pore size distribution data is given in Supplementary Figure 6 in the Supplementary Information.

REVIEWERS' COMMENTS:

Reviewer #1 (Remarks to the Author):

I am satisfied with the authors's response to my questions and would recommend for publication.

Reviewer #2 (Remarks to the Author):

Li and co-authors proposed a new concept of membrane manufacturing called combined crystallization and diffusion (CCD) method. The authors had revised the manuscript on the basis of the reviewer's comments. However, some questions are considered as follows:

1. Supplementary Figure 6: It is better to figure out more precise data and curves of pore size distribution curves for a clear review.
2. Which fields could be CCD membranes applied for? If there are some practice use or tests for separate or purify some specific solution, display them to prove the performances of CCD membranes.

Therefor, the present manuscript could be published after minor revision.

Reviewer #3 (Remarks to the Author):

My comments have been addressed properly.

Response to the Referees

Reviewer #1 (Remarks to the Author):

I am satisfied with the authors's response to my questions and would recommend for publication.

Reviewer #2 (Remarks to the Author):

Li and co-authors proposed a new concept of membrane manufacturing called combined crystallization and diffusion (CCD) method. The authors had revised the manuscript on the basis of the reviewer's comments. However, some questions are considered as follows:

1. Supplementary Figure 6: It is better to figure out more precise data and curves of pore size distribution curves for a clear review.

Response: we have given values of mean flow pore size and pore size range (biggest and smallest pore sizes) in the figure legend (Supplementary Fig.3 in the revised Supplementary Information). The

curves have already given full information of the pore size distribution, but we agree with the reviewer that precise values would make readers easier to catch important characteristics.

2. Which fields could be CCD membranes applied for? If there are some practice use or tests for separate or purify some specific solution, display them to prove the performances of CCD membranes.

Response: we have specified the applications in the introduction part of the paper that the membranes are for filtration purposes, and they include a wide range of applications wherever a subject is bigger than the pore size of the membrane and it can be retained by the membranes. The CCD membranes can be used in but not limited to drinking water production, wastewater treatment, dialysis, beverage clarification, etc. In our study, we used the CCD Al/Al membranes to purify a suspension of ~50 nm particles and the membranes could extract clear water from the suspension, which demonstrates the effectiveness of the membrane. But applying membranes in a specific filtration process is a more target-centred topic and is affected by complex operating factors, although it would be interesting, we feel it should be another separate research but not a necessary part of the current topic of membrane science.

Therefore, the present manuscript could be published after minor revision.

Reviewer #3 (Remarks to the Author):

My comments have been addressed properly.